## Perspective

policy change; human rights; aging; law; mental health

**Corresponding author:**
Sarah Steele;
Email: ss775@cam.ac.uk

# Pension reforms, economic security, and mental health: The need for a human rights-based approach

Sarah Steele[1,2,3] 🔟, Milagros Ruiz[1,4], Matthew Parbst[3] and David Stuckler[3]

[1]School of Health and Social Care, University of Essex, Colchester, UK; [2]Intellectual Forum, Jesus College, Cambridge, UK; [3]Dondena Centre, Bocconi University, Milan, Italy and [4]Research Department of Epidemiology and Public Health, University College London, London, UK

## Abstract

Pension systems play a crucial role in providing economic security and supporting well-being in later life. However, as governments implement reforms to ensure financial sustainability—such as raising the retirement age, reducing benefits, and shifting to defined-contribution schemes—these measures often overlook their psychological and social consequences. Pension insecurity has been linked to heightened stress, anxiety, and depression, as well as increased social isolation, particularly among vulnerable populations, including those in physically demanding jobs, low-income workers, and individuals with existing health conditions. Despite clear evidence of these effects, mainstream pension reform discourse prioritises fiscal concerns over social and mental health implications. This article examines pension reform through the Human Rights–Public Health Pension Framework (HRPHPF), integrating legal, public health, and policy perspectives to assess its impact on mental well-being. It situates pension rights within international human rights law, explores the psychological risks associated with pension insecurity, and advocates for a human rights-based approach to pension policymaking. The article calls for integrating mental health impact assessments into pension reforms to prevent adverse outcomes and ensure that policies promote dignity, social inclusion, and economic security in old age. A more balanced approach is necessary to align financial sustainability with broader well-being and human rights principles.

## Impact statement

As populations age and governments confront fiscal constraints, pension reform is rapidly becoming a defining policy issue of the 21st century. Yet, the profound psychological and social effects of these reforms remain largely invisible in mainstream debates. This article brings to light how pension insecurity and extended working lives can undermine mental health, particularly among older adults in precarious work, those facing discrimination, and individuals already managing physical or psychological vulnerabilities. By framing pension reform as not only an economic or actuarial concern but also a matter of public health and human rights, this work helps shift the discourse toward more humane and socially inclusive policymaking. Our research presents a unique, interdisciplinary framework—the Human Rights–Public Health Pension Framework (HRPHPF)—that policymakers, human rights advocates, and public health professionals can utilise to assess the mental health and equity implications of pension changes. This framework has the potential to influence national reform agendas and encourage international organisations to set clearer mental health standards in social protection policy design. The work also highlights the gendered and global dimensions of pension reform, calling attention to the risks faced by older women and people in countries without formal pension systems. The article's recommendations—including the use of mental health impact assessments and flexible retirement models—are directly actionable and relevant to ongoing reforms in both high- and low-income countries. By connecting economic policymaking with mental health and rights-based governance, this research supports the development of more resilient, inclusive, and ethical ageing policies. Ultimately, it helps ensure that pension reforms promote dignity and well-being, not just fiscal balance, for current and future generations.

## Introduction

Pension systems constitute a foundational instrument of social protection, ensuring economic security and quality of life for individuals as they transition out of the workforce due to age or other reasons. However, in response to increasing longevity, declining fertility rates, and mounting fiscal pressures, governments worldwide have introduced a series of pension reforms

to cut costs. These policy shifts—ranging from raising the retirement age to reducing state-funded benefits and shifting from defined-benefit to defined-contribution schemes (Grech, 2018)—are often framed as necessary adaptations to contemporary economic and demographic challenges (Barr, 2013). Often part of broader neoliberal economic policy shifts promoted by international financial institutions, which prioritise fiscal consolidation and market-driven models of welfare provision, these reforms frequently fail to acknowledge the profound implications of pension reforms for mental health.

A growing body of research has established the intricate relationship between financial stability and psychological well-being in later life. Pension insecurity has been shown to exacerbate stress, anxiety, and depression (Kentikelenis et al., 2011), while also increasing the risk of social isolation, a key determinant of mental health decline (Shultz and Wang, 2011; Lowies et al., 2021; Jhuremalani et al., 2022; Bosco et al., 2024). Policies that delay retirement or reduce pension adequacy disproportionately affect individuals engaged in physically demanding labour, those with limited employment alternatives, and those already experiencing health vulnerabilities (van der Heide et al., 2013). Despite these well-documented effects, mainstream pension reform discourse remains primarily preoccupied with fiscal sustainability, often treating social and psychological considerations as secondary or extraneous concerns.

This article critically examines pension reform through the lens of the Human Rights–Public Health Pension Framework (HRPHPF), which integrates legal, public health, and policy analyses to assess how pension restructuring impacts mental health. First, the discussion situates pension rights within the broader context of international human rights law, demonstrating how existing legal frameworks establish pension security as a fundamental entitlement rather than a discretionary welfare provision (United Nations, 1966). Second, it examines the public health implications of pension insecurity, drawing on empirical research that suggests people facing economic precarity in old age are particularly vulnerable to poor mental health (Marshall et al., 2021; Choi et al., 2023). Finally, it advocates for a human rights-based approach to pension policymaking, arguing that mental health impact assessments should become an integral component of pension reform processes to ensure that policy decisions do not disproportionately disadvantage older individuals. We argue for a shift in pension reform discourse from purely fiscal concerns to a broader framework that prioritises well-being. While this article primarily focuses on pension reforms in countries with established systems, it is vital to acknowledge that most of humanity, lack comprehensive pension coverage. In these contexts, retirement is often not a practical reality, and older individuals remain in informal labour or rely on kinship support networks (ILO, 2018). Thus, global pension reform debates must be attentive to structural inequalities and the absence of formal security mechanisms. This article, thereby, highlights the legal, health, and social dimensions of pension reform and contributes to a growing discourse on the intersection of social protection, mental health, and human rights.

## The foundations of pension rights in international human rights law

The right to social security, including pensions, is enshrined in international human rights law and forms a cornerstone of social protection systems worldwide. Legal instruments, such as the *Universal Declaration of Human Rights* (UDHR) and the *International Covenant on Economic, Social and Cultural Rights* (ICESCR), explicitly recognise the obligation of states to ensure financial security for older adults (United Nations, 1948, 1966). These instruments establish pensions not as discretionary welfare benefits but as fundamental entitlements that states must protect to ensure dignity, equality, and social inclusion. However, as governments introduce pension reforms in response to economic pressures, concerns arise that these changes are eroding protections that states are legally bound to uphold.

Notably, the ICESCR explicitly recognises pension rights. Article 9 guarantees "the right of everyone to social security, including social insurance," while Article 11 affirms the right to an adequate standard of living. These provisions are reinforced by *General Comment No. 19* of the UN Committee on Economic, Social and Cultural Rights (CESCR), which clarifies that states must ensure their pension systems are adequate, accessible, and sustainable (CESCR, 2008). Adequacy requires pension benefits to be sufficient to maintain a decent standard of living. Accessibility mandates that pension schemes cover all individuals, particularly those in vulnerable groups, and sustainability ensures that pension systems remain financially viable without progressively weakening protections. The CESCR also highlights the principle of non-retrogression, which prohibits states from reducing social security protections unless such measures are unavoidable, temporary, and justified by exceptional circumstances (CESCR, 2008). This principle is particularly relevant as many states implement austerity-driven pension reforms that reduce benefits without adequate safeguards.

Judicial bodies and human rights institutions have reinforced these obligations in various judgements. The European Court of Human Rights (ECtHR), for example, has ruled in multiple cases that pension rights, while subject to state regulation, must not be arbitrarily reduced in a way that disproportionately affects individuals' standard of living. In *Kjartan Ásmundsson v. Iceland* (European Court of Human Rights, 2004), the ECtHR held that the complete revocation of a disability pension violated Article 1 of Protocol No. 1 of the *European Convention on Human Rights* (ECHR), which protects the right to property (ECtHR, 2004). The court emphasised that while states retain discretion in social security matters, pension entitlements, once granted, constitute a form of property that cannot be arbitrarily revoked. Similarly, in *Béláné Nagy v. Hungary* (European Court of Human Rights, 2016), the ECtHR ruled that Hungary's decision to withdraw a disability pension benefit without transitional measures violated the right to property, reaffirming that pension rights must be subject to legal certainty and foreseeability (ECtHR, 2016).

The Inter-American Court of Human Rights (IACtHR) has also recognised pension rights as fundamental. In *Five Pensioners v. Peru* (Inter-American Court of Human Rights, 2003), the court ruled that the Peruvian government's decision to cut pension benefits violated the right to judicial protection (Article 25) and the right to property (Article 21) of the *American Convention on Human Rights* (ACHR) (IACHR, 2003). The ruling established that pension reductions that disproportionately impact retirees without sufficient legal safeguards or justification can constitute a human rights violation. The Inter-American Commission on Human Rights (IACHR) has also emphasised that pensions are a key mechanism for preventing poverty and ensuring dignity for older persons, reinforcing that governments must not implement pension reforms that disproportionately harm vulnerable populations (Numi, 2021).

The International Labour Organisation (ILO) further strengthens the legal framework for pension rights. The *Social Protection*

*Floors Recommendation* (No. 202) establishes that states should ensure "older persons have income security at least at a nationally defined minimum level" (ILO, 2012). This recommendation supports the progressive realisation principle, which requires states to improve pension protections rather than continuously weakening them over time. Despite financial challenges, states must work toward expanding pension coverage and adequacy rather than enacting regressive reforms that leave retirees without sufficient income. The ILO (2025) highlights that privatising pensions and increased reliance on defined-contribution schemes often exacerbate financial insecurity, particularly for women and low-income workers, and must be cautiously approached.

Regional human rights instruments further reinforce these principles. The African Charter on Human and Peoples' Rights (ACHPR) recognises the right to social security as essential for economic and social well-being. The African Commission on Human and Peoples' Rights has ruled in multiple cases that states must ensure adequate pension protections and prevent discrimination in access to social security benefits. The *European Social Charter*, enforced by the European Committee of Social Rights (ECSR), establishes pension adequacy as a fundamental requirement, with the ECSR ruling against states that fail to provide sufficient old-age benefits (Council of Europe, 2025).

Governments implementing pension reforms must also ensure transparency, accountability, and participation in decision-making. The ICESCR and related human rights instruments emphasise that individuals must have mechanisms to challenge pension reductions or exclusions, including judicial review, complaint mechanisms, and participatory decision-making processes (Rodriguez-Pinzon & Martin, 2002). Courts in multiple jurisdictions have reinforced this right. The German Federal Constitutional Court, for example, ruled in *BVerfG 1 BvL 1/09* (2010) that pension reductions must be subject to legal predictability and transitional measures to prevent economic hardship among retirees. The South African Constitutional Court has similarly ruled that pension rights must be protected as part of the constitutional right to social security, as seen in *Khosa v. Minister of Social Development* (2004), which established that non-citizens could not be arbitrarily excluded from pension benefits.

Taken together, these legal decisions and treaties establish a human rights basis for pension policies not just to be reformed to be financially sustainable but to require that states pursue reforms that do not erode protections that guarantee dignity for older people. When pension reforms reduce coverage, adequacy, or accessibility, states may violate human rights obligations. The legal principles outlined in international treaties, judicial rulings, and human rights recommendations provide a comprehensive foundation for evaluating pension reforms not merely as economic adjustments but as matters of fundamental rights and social justice. As pension systems evolve in response to demographic and financial challenges, adherence to these legal standards remains essential to ensuring that reforms do not disproportionately harm those relying on pensions for financial security.

## Pension reforms and their psychological impact

Economic security is a fundamental determinant of mental health in later life, yet pension reforms often generate financial instability not through reductions in nominal entitlements, but through uncertainty around the adequacy of inflation indexation. When pensions are fixed in nominal terms at retirement, their real value depends on future price levels, making insufficient or uncertain indexation a key driver of anxiety and psychological distress among pensioners, particularly in inflationary contexts where purchasing power can erode substantially over time. Financial insecurity is a significant stressor, particularly for older individuals who lack alternative sources of income (Bagnall and Harris, 2021). Reports from Australia indicate that the fear of outliving one's savings and the inability to afford basic necessities contribute to financial stress in older adults (Jhuremalani et al., 2022), which further deteriorates their psychological well-being (Lowies et al., 2021). The implications of this reality for pension systems are exceedingly important, and reforms have been shown to have profound and direct effects on mental health after retirement. For example, a natural experiment study linked the 2006 reform to reduce the replacement rate of the Dutch pension system with increased economic uncertainty and depression among retirees (Grip et al., 2012). Contrarily, the South African reform to lower the age eligibility criteria for a pension payout from 65 to 60 years of age in 2008 reduced the risk of traumatic stress and depression among male pensioners (Mostert et al., 2022). This quasi-experimental evidence emphasises the diverging and unintended consequences of these policy changes, which can now be viewed as a health-adverse reform on one hand versus a health-promoting reform on the other.

Extended working life, a policy response to economic pressures, also imposes significant mental health burdens on older workers. Raising the retirement age is often justified by increases in life expectancy, yet this rationale fails to account for disparities in health, job roles, and conditions of employment. While some individuals with prolonged working lives appear to age more successfully, others, particularly those in stressful, precarious, or physically demanding jobs, face heightened levels of social isolation (Nilson et al., 2022), and depression during retirement (Wahrendorf et al., 2013). The UK Government Office for Science (2019) has emphasised that these challenges are exacerbated by age discrimination as older workers may struggle to secure employment and maintain financial stability. In turn, age discrimination is known to lower subjective well-being and increase psychological distress (Kang and Kim, 2022). The expectation that individuals will work longer ignores the reality that many older adults cannot do so due to health reasons, limited job opportunities, or unpaid care responsibilities.

The psychological consequences of pension insecurity extend beyond financial concerns to encompass broader issues of identity and social integration. Retirement represents a significant life transition, and sudden pension reforms can disrupt an individual's sense of purpose, leading to feelings of worthlessness and social withdrawal (Shultz and Wang, 2011). Many individuals derive their sense of identity and self-worth from their professional roles, and abrupt changes to pension policies can leave retirees feeling displaced and undervalued (Shultz and Wang, 2011; Bosco et al., 2024). Financial instability not only threatens the mental health of older adults (Marshall et al., 2021; Choi et al., 2023) but also further limits their social participation, as they may be forced to reduce or eliminate activities that provide a sense of community and belonging (Jhuremalani et al., 2022). The intersection of economic insecurity and social isolation creates a feedback loop that exacerbates mental health issues, reinforcing the need for pension policies that support both financial stability and social well-being.

Gender disparities in pension security highlight the uneven mental health burden of pension reforms. Women, on average, have lower lifetime earnings and are more likely to rely on non-contributory pensions due to career interruptions related to caregiving responsibilities (Zaidi and Mirza-Davies, 2024). Reductions in state pension benefits disproportionately affect older women

(Zaidi and Mirza-Davies, 2024), and particularly those from racially marginalised groups (Hogan and Perrucci, 2007). Women's greater burden of financial insecurity has been linked with higher levels of psychological distress (Kirkman and Fisher, 2021). Yet, reforms, including changes to the state pension age, are a particular concern for women due to their longer life expectancy (Government Office for Science, 2019). The gendered nature of pension policy is rarely acknowledged in reform debates, yet it is a critical factor in understanding the broader social implications of changes to pension systems. Addressing these disparities requires policies that recognise and mitigate the unique vulnerabilities faced by women in later life.

While this article highlights old-age economic insecurity, the broader human right at stake is economic security throughout life. Article 25 of the UDHR affirms a right to protection against unemployment, sickness, disability, and widowhood. Mental and physical health impacts of economic insecurity extend beyond retirees, affecting entire populations during economic downturns or welfare retrenchment. Notably, scholars such as Watson and Osberg (2017) and Rohde et al. (2020) emphasise the psychological consequences of sustained economic insecurity, which is distinct from material deprivation. They highlight how the persistent threat of income loss or inadequate social protection can adversely affect mental health across the life course.

## Future perspectives: A human rights-based approach to pension reform

Future pension reform must move beyond a narrow fiscal focus and embrace a more integrated, rights-based approach that considers reforms' legal, economic, and mental health implications (Sepúlveda and Nyst, 2012). As pension systems evolve in response to demographic pressures and financial constraints, it is imperative to ensure that reforms do not disproportionately harm vulnerable populations. The increasing reliance on defined-contribution schemes, the gradual erosion of state-backed pensions, and the steady rise in retirement age pose significant risks to financial security and psychological well-being in later life. These shifts, often justified on economic grounds, frequently overlook states' broader human rights obligations under international law, including those outlined in the ICESCR and the UDHR. While these legal instruments affirm the right to social security, enforcement mechanisms remain weak, and many states continue to implement austerity-driven pension reforms without adequate safeguards.

A more robust legal framework is necessary to ensure that pension rights are not subject to arbitrary reductions undermining financial security in old age. Legal challenges underscore the crucial role of judicial oversight in safeguarding pension entitlements. Strengthening legal protections at the national level through constitutional amendments and statutory rights would provide a more stable foundation for pension rights, preventing regressive policy shifts that disproportionately impact low-income workers, women, and those in precarious employment. As articulated by the CESCR, the principle of non-retrogression must be given greater weight in policy decisions, ensuring that states cannot reduce pension benefits without demonstrating necessity, proportionality, and the absence of viable alternatives. Furthermore, international bodies such as the ECSR and the IACtHR should be more proactive in holding states accountable for pension-related human rights violations. Future reforms should also draw on the United Nations Guiding Principles on Human Rights Impact Assessments of Economic Reforms (A/HRC/40/57), which offer a blueprint for assessing reforms in a manner consistent with international human rights law.

Beyond legal protections, pension reforms must incorporate public health considerations, particularly given the growing evidence linking pension insecurity to adverse mental health outcomes. Economic instability in later life is a well-documented risk factor for anxiety, depression, and social isolation (Marshall et al., 2021; Choi et al., 2023), and especially for older adults residing in countries without generous social protection schemes (Sjöberg, 2023). Therefore, pension reforms that delay retirement or reduce entitlements exacerbate these risks (Grip et al., 2012). Those in physically demanding jobs or precarious employment are particularly vulnerable to the effects of extended working life (Wahrendorf et al., 2013; Nilson et al., 2022), as declining health may make continued employment untenable. Yet, pension policy discussions rarely account for the psychological dimensions of financial insecurity. Therefore, mental health impact assessments should be embedded into pension policymaking, functioning as a critical tool to evaluate how proposed reforms affect psychological well-being, social participation, and overall quality of life.

A more flexible approach to retirement could also mitigate some of the adverse effects of pension insecurity. Phased retirement models, which allow workers to transition gradually from full-time to part-time work before fully retiring, have reduced stress levels while ensuring greater financial stability. Countries that successfully implement phased retirement systems, such as Sweden and the Netherlands, provide helpful case studies for future policy development (Lachowska et al., 2009; Bloemen et al., 2016). Additionally, non-contributory pension schemes should be expanded, particularly for those who experience interruptions in their working life, such as women with caregiving responsibilities. Ensuring greater inclusivity in pension systems will help prevent long-term economic disparities that disproportionately impact certain demographics.

International collaboration on pension reform standards will also be instrumental in ensuring that future pension systems align with principles of equity, sustainability, and well-being. Organisations such as the UN, ILO, and regional human rights bodies should play a greater role in establishing global benchmarks for pension policies, ensuring that economic efficiency is balanced with social justice considerations. Comparative research across different pension models will be essential in identifying best practices for safeguarding dignity in old age while addressing the mental health challenges associated with economic insecurity. Establishing a global framework for pension adequacy that integrates legal, economic, and health-based standards will be crucial for the long-term sustainability of pension systems.

As pension systems evolve, a rights-based approach prioritising economic security, mental health, and social inclusion will be fundamental to ensuring fairness and sustainability. Future pension reforms can move beyond a narrow fiscal focus by integrating legal protections, mental health impact assessments, flexible retirement options, and international collaboration, fostering inclusive, resilient, and socially responsible approaches to ageing and retirement security.

**Open peer review.** To view the open peer review materials for this article, please visit http://doi.org/10.1017/gmh.2025.10047.

**Author contribution.** SS conceived the study. All authors contributed to the drafting and critical revision of the manuscript and approved the final version for submission.

**Financial support.** This research was supported by a European Research Council Investigator Award.

**Competing interests.** SS, MR, and MP have nothing to declare. DS is funded by a European Research Council Investigator Award and has no conflict of interest to declare.

**Ethical statement.** This study did not require ethical approval as it did not involve human participants, human data, or human tissue.

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
