## [Reviewer Report]

The article addresses a very timely and important problem, as the human rights (mental health, specifically) implications of pension reforms.

I think the article lacks a contextualization of the pension reforms, which usually take place in the frame of economic orthodox/neoliberal reforms, which are mainly (not only) promoted by int. financial institutions.

To complete the normative framework of a Human Rights-Based Approach to pension reforms, I suggest to have a look at the UN Guiding principles on human rights impact assessments of economic reforms (A/HRC/40/57) https://www.ohchr.org/en/documents/thematic-reports/ahrc4057-guiding-principles-human-rights-impact-assessments-economic

---

## [Reviewer Report]

Review of

“Pension Reforms, Economic Security, and Mental Health: The Need for a Human Rights-Based Approach.”

Lars Osberg, Dalhousie University

This is a good article, which could be a better article if some of its arguments were made more effectively.

1. The paper starts with the observation that “as governments implement reforms to ensure financial sustainability……” – but the term “sustainability” escapes scrutiny. In general, an expense is sustainable if people are now willing to pay it and continue to be willing to pay it. Everywhere in the world the birth rate has decreased in recent years, often with dramatic rapidity, which necessarily implies shifts in the age distribution of the population and ongoing changes in the ratio between people of “retirement” age and younger cohorts. What exactly does “sustainability” mean in this context? It is inescapable that falling birth rates will mean there are ever fewer middle-aged people for each elderly person, so the middle-aged will each have to pay a little more to support a given level of consumption for their parents and their parents’ peers, but if a relative scarcity of middle aged workers implies higher wages, in what sense is that “unsustainable”?

2. Although the article is phrased quite broadly, the implicit context appears to be rich, developed countries in which pension systems already exist for the retirement years of the vast majority of the population, who have been in paid employment throughout their working lives. Particularly in sub-Saharan Africa, however, many countries do not now have in place pension systems which pay benefits to an appreciable percentage of the population. Without pensions, “retirement” is not an operative concept for most people and in much of the world, the elderly typically continue to work in subsistence agriculture or the informal urban economy as long as they are able and have to depend on shared consumption within extended family settings for survival. Some recognition of the non-existence of pension systems for a very large percentage of the world’s population would be useful.

3. The article’s emphasis on the mental health impacts of inadequate pensions for the elderly from a human rights perspective is a subset of the more general impacts of economic insecurity on health and the basic human right to economic security.

Article 25: of the Universal Declaration of Human Rights states:

“Everyone has the right to a standard of living adequate for the health and well-being of himself and of his family, including food, clothing, housing and medical care and necessary social services and the right to security in the event of unemployment, sickness, disability, widowhood, old age or other lack of livelihood in circumstances beyond his control.” United Nations (1948)

A right to security in “Old Age” is clearly specified, but so also is the right to security in the event of “unemployment, sickness, disability, widowhood or other lack of livelihood in circumstances beyond his control.”

Since a substantial literature has documented the health implications – physical and mental- of economic insecurity, it would strengthen the article to note that its focus on the mental health implications of the lack of delivery on the human right to an adequate pension in old age is only a part of the implications of economic insecurity – and that the basic human right to economic security is not limited to the elderly.

“Blown Off-Course? Weight Gain Among the Economically Insecure During the Great Recession” Angela Daley, Nick Rohde, Lars Osberg and Barry Watson Journal of Economic Psychology Volume 80, October 2020, 102289 https://doi.org/10.1016/j.joep.2020.102289

-“Welfare-Based Income Insecurity in the US and Germany: Evidence from Harmonized Panel Data” Nicholas Rohde, Kam Ki Tang, Conchita D’Ambrosio, Lars Osberg and Prasada Rao Journal of Economic Behavior and Organization, Volume 176, August 2020, Pages 226-243

-“The Shattered “Iron Rice Bowl”: Intergenerational Effects of Chinese State-Owned Enterprise Reform” Nancy Kong, Lars Osberg and Weina Zhou Journal of Health Economics Volume 67, September 2019, 102220 https://www.sciencedirect.com/science/article/pii/S0167629618309664

“Job Insecurity and Mental Health in Canada” Barry Watson and Lars Osberg Applied Economics published online February 21,2018 http://www.tandfonline.com/doi/full/10.1080/00036846.2018.1441516

“Healing and/or Breaking? The Mental Health Implications of Repeated Economic Insecurity” Lars Osberg and Barry Watson Social Science and Medicine Volume 188, September 2017, Pages 119-127

“Is It Vulnerability or Economic Insecurity that matters for Health?” Nick Rohde, K.K. Tang, Lars Osberg and Prasada Rao Journal of Economic Behavior & Organization 2017, 134: 307-319

“The Self-Reinforcing Dynamics of Economic Insecurity and Obesity” Nick Rohde, Lars Osberg and K.K.Tang Applied Economics 49:17 1668-1678 November, 2016

“Economic Insecurity and the Weight Gain of Canadian Adults: A Natural Experiment Approach” Barry Watson, Lars Osberg and Shelley Phipps, Canadian Public Policy June 2016 Pp.115-131,

“The Effect of Economic Insecurity on Mental Health: Recent Evidence from Australian Panel Data” Nick Rohde, Prasada Rao Lars Osberg and K.K. Tang (2016) Social Science & Medicine Vol.151, February 2016, Pp. 250-258

Economic Insecurity and Well-Being Lars Osberg (2021) Department of Economic and Social Affairs, United Nations, DESA Working Paper (2021) https://www.un.org/sites/un2.un.org/files/wp173_2021.pdf

---

## [Editor Report]

Dear authors,

The reviewers have suggested some minor revisions to your manuscript. Please respond to the reviewers' suggested changes to the best of your ability and revise your manuscript.

Once again, thank you for submitting your manuscript to Cambridge Prisms: Global Mental Health and I look forward to receiving your revised version.

Sincerely,

Dr. Maria Francesca Moro

---

## [Reviewer Report]

Page 3:

“a series of pension reforms to drive financial sustainability.”

More accurate to say “a series of pension reforms to cut costs”. Financial sustainability depends on costs and on willingness to pay (which is not at all discussed, or influenced, by these reforms)

Page 4

“at many low- and middle-income countries, especially in sub-Saharan Africa,”

more accurate to say “for most of humanity”

– the way this is now phrased, it sounds as if the authors want to say that “sometimes” the actual availability now of adequate pensions is inadequate – which glosses over the fact that where there is some formal entitlement for the indigent elderly (e.g. India and China), it is at very low rates of payment.

Page 9

“Economic security is a fundamental determinant of mental health in later life, yet pension reforms frequently introduce financial instability that exacerbates mental health vulnerability.”

When pension entitlement is assessed at the time of retirement and pension claim, it is fixed in nominal terms but the price level determines its real value. Anxiety and financial instability for pensioners is rarely about the possibility of downward revisions in the nominal value of their pension – but there is often anxiety and insecurity about whether the pension will be adequately indexed for inflation, which can erode the real value of the pension substantially (e.g. 5% inflation over 20 years means a nominal pension loses 62% of its value). In this article there is no mention of inflation or of the appropriate method of indexation of pensions to account for inflation – but the whole argument about insecurity / anxiety / mental health impacts depends on uncertain indexation in an inflationary context.

Some discussion of inflation and indexation is needed.

Page 21

Watson, B., & Osberg, L. (2017). Healing and/or breaking? The mental health implications of repeated economic security. Social Science & Medicine, 188:119 127. https://doi.org/10.1016/j.socscimed.2017.06.042

Watson & Osberg were writing about insecurity

Watson, B., & Osberg, L. (2017). Healing and/or breaking? The mental health implications of repeated economic insecurity. Social Science & Medicine, 188:119 127. https://doi.org/10.1016/j.socscimed.2017.06.042

---

## [Editor Report]

Thank you for the thoughtful revisions, they have greatly strengthened the manuscript. Reviewer 2 has suggested a few minor final revisions which would further clarify the paper.

---

## [Reviewer Report]

The article is a useful reminder of the practical implications of human rights. Only one more suggestion - please correct the title of Watson and Osberg in the references - it is

Watson, B., & Osberg, L. (2017). Healing and/or breaking? The mental health

implications of repeated economic insecurity

---

## [Editor Report]

- please correct the title of Watson and Osberg in the references - it is

Watson, B., & Osberg, L. (2017). Healing and/or breaking? The mental health

implications of repeated economic insecurity